# Testing and Analysis Fault of Induction Motor for Case Study Misalignment Installation Using Current Signal with Energy Coefficient

Supachai Prainetr, Satean Tunyasrirut and Santi Wangnipparnto *

Department of Electrical Engineering, Faculty of Engineering, Pathumwan Institute of Technology, Bangkok 10330, Thailand; prainetr@gmail.com (S.P.); satean2000@gmail.com (S.T.)
* Correspondence: nipparnto@gmail.com; Tel.: +66-818148671

**Abstract:** An induction motor is a key device for an industrial machine. The installation misalignment of the motor will result in derating problems and energy consumption that is generally used to analyze signal faults using the fast Fourier transform (FFT) method. Problems with the rotor affect the non-stationary signal and FFT can be utilized to analyze this problem inefficiently. This paper proposed the testing and analysis of faults in an eccentric rotor at different levels using the stator current detection technique and the calculation of the energy signal coefficient via the wavelet decomposition (WD) method. The experimental results showed that an increase in eccentricity had a linear relation with the energy signal, where $R^2$ was 80.81%. Moreover, the test results illustrated that the proposed method was more efficient than FFT and applicable to motor fault analysis and application in the industrial.

**Keywords:** induction motor; misalignment installation; stator current signal; energy coefficient

## 1. Introduction

An induction motor is an essential component of industrial machinery. However, its long service period and high-temperature environment lead to its deterioration. Induction motor malfunctions are classified into two types—mechanical malfunctions include eccentric rotor shafts, rotor cracks, and bearing cracks, while electrical malfunctions include shorted stator coils and unstable motor voltage. Statistically, malfunctions are most commonly due to [1–3] bearing faults (40%), followed by stator winding faults (38%), rotor faults (10%), and other parts (12%) [4–6]. The data reveal that problems frequently occur in the bearing because the structure of the rotor motor is designed to directly bear the load from the motor shaft. Bearing damage is often a side-effect of a fault in the eccentric shaft, which has many possible causes, such as inaccurate assembly of the motor at the manufacturer or imbalanced installation that creates an eccentric air gap between the rotor and stator, in addition to vibration [7]. Therefore, the eccentric problem is one of the main faults leading to other malfunctions. There are three categories of eccentric air gaps in the motor—static, dynamic, and hybrid. Moreover, the imbalance of the motor shaft or distortion of the motor structure may create noise during rotation, while the rotor axis or bearing distortion creates an asymmetric air gap, leading to overheating of the coil, misaligned coupling, a tight belt between the axis, and an unstable or imbalanced motor base attachment.

Recently, research has been carried out to establish methods for the analysis of eccentric induction motor malfunctions by using signal data. These signal-based analyses are classified into the following four methods:

1.  Analysis of an electric current signal while the motor is running;
2.  Analysis of a vibration signal while the motor is running;
3.  Analysis of a sound signal;

4.      Analysis of thermal imagery.

According to a literature review of research conducted from 1975 to 2017, a number of studies have investigated induction motor malfunctions [8–10]. For instance, S. Nadi et al. proposed a technique for the inspection of mechanical malfunctions in an induction motor using the relation between the eccentric motor and harmonics. Similarly, Thomson et al. proposed an analysis technique for mechanical malfunctions using the current signal to analyze the frequency components of the current and vibration signals by applying the fast Fourier transform (FFT) method. However, FFT is not capable of analyzing the resolution of non-stationary, unstable signals that occur in the transient condition. Subsequently, a signal analysis method based on wavelets was developed, which can analyze non-stationary signals and identify the required frequency range [10–12]. Moreover, the wavelet extracts the signal data for use in the analysis of artificial intelligence (AI) systems, in addition to other systems such as neural, fuzzy logic, and neural fuzzy networks [13,14]. The major components of a condition monitoring system include the machinery, condition monitoring sensors, signal processors, fault classifiers, machine models, and the monitoring output. Errors and uncertainties in fault classification can lead to false alarms, which necessitate better, more robust, and more reliable condition monitoring systems. Moreover, a major challenge for a condition monitoring technique is its ability to differentiate changes in the signal that are due to machinery defects. The proposed statistical method is based on the identification of differences in operating conditions that reach statistical significance.

This paper proposes the testing and analysis of faults caused by misalignment due to the incorrect installation of an induction motor. The performance of the proposed method was tested by first assessing the motor in its normal state and coupling it with a load, and then experiments were carried out in the abnormal condition. The hypothesis of this study is that the external misalignment force leads to changes in the magnetomotive force (MMF) and permeance wave, which results in a non-stationary signal [15–17]. FFT is not appropriate for signal frequency detection in transient condition [18,19]. Therefore, this research applied the wavelet transform with multiple levels of resolution distribution. The energy coefficient with its standard deviation can be extracted using the wavelet transform [20,21]. In order to obtain the attributes of the abnormal frequency signal from the external misalignment, experiments at 10%, 20%, 30%, and 40% misalignment levels (relative to the standard alignment) were used to perform a data correlation analysis and diagnose rotor misalignment effects due to incorrect installation. This method can be used in preventive maintenance planning to avoid motor deterioration, especially for larger motors that cannot be stopped.

## 2. The Conception and Proposed Methods

### 2.1. Installation Misalignment Types

Shaft misalignment is the most common factor in the damage of an induction motor in a machine. Misalignments can be parallel, angular, or a combination of both parallel and angular, as shown in Figure 1.

### 2.2. Digital Signal Processing Techniques

The method for detecting mechanical faults due to misalignment of the rotor shaft is detailed below. The fault can be analyzed by measuring the frequency of the signal, detecting the motor current signal, and calculating the frequency from the following equation:

$$f_{ec,i} = f_s(1 \pm k\frac{1-s}{p}) \tag{1}$$

where $f_{ec,i}$ is the frequency due to the installed rotor misalignment, $f_s$ is the fundamental frequency, k is the order number, s is the slip speed, and p is the number of magnet poles.

$$f_{ec,p} = f_s k(\frac{1-s}{p}) \tag{2}$$

where $f_{ec,p}$ is the frequency of the rotor misalignment with referring to the centerline to analyze the relation and impacts of the eccentric rotor.

From Equations (1) and (2), the fault can be characterized by monitoring the frequency modulation from the electrical supply and the stator current to identify broadband changes in the stator current frequency.

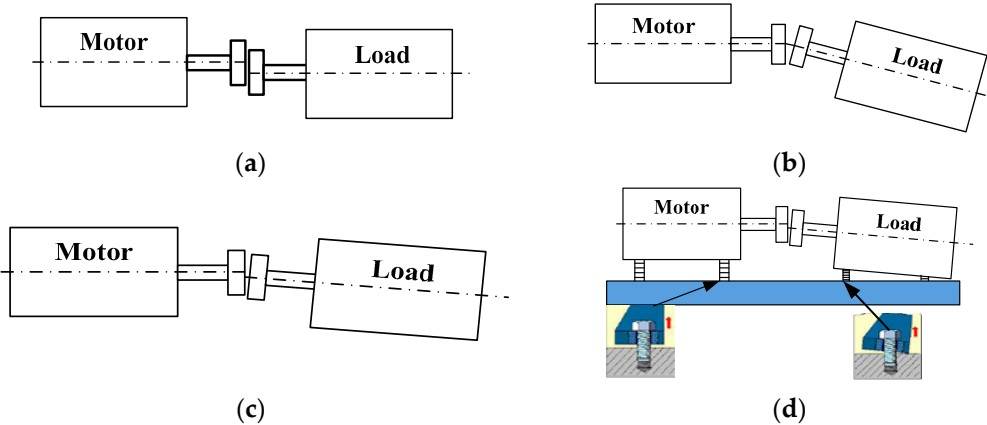

**Figure 1.** The incorrect installation of an induction motor. (**a**) shows a parallel misalignment arising from the motor shaft axis connected to the load in the same plane, and the center of motion of the shaft is parallel. (**b**) shows an angular misalignment arising from the axial motor shaft coupled with the load in the same plane, and the center of motion of the shaft is angular. (**c**) shows a combined misalignment arising from the axial motor shaft coupled with the load in the same plane, and the center of motion is parallel and angular. (**d**) shows the adjustment of the installed rotor shaft connected to a load calibrated with a dial gauge.

### 2.3. Analysis Technique and Determine Energy Coefficient of Current Signal

The wavelet transform (WT) method is a technique developed from the short-time Fourier transform (STFT), which uses the window size function modification principle (window function). The time interval must be suitable for the frequency range to be analyzed; the higher the frequency signal is, the shorter the analysis time. Thus, lower-frequency signals have a wide time interval. For the wavelet transform, the concept of multi-resolution analysis is used by converting the signal into small waves with limited energy. It is therefore suitable for the analysis of transient current signals in the frequency domain using wavelet separation. This study applied the frequency band energy ratio in [22], which is calculated using Equations (3)–(5).

According to the energy conservation principle, the following relation is obtained:

$$E_n(x(t)) = \sum_{m=0}^{2^k-1} E_n(x^{k,m}(i)) \tag{3}$$

where the proportion of the mth frequency band energy relative to the total energy, i.e., the normalized frequency band ratio, is

$$E_n(m) = \frac{E_n(x^{k,m}(i))}{E_n(x(t))} \tag{4}$$

and the sum of all frequency band energy ratios is equal to 1, i.e.,

$$\sum_{m=0}^{2^k-1} E_n(m) = 1 \tag{5}$$

### 3. Research Methods

This article reports the results of laboratory tests in which fault signals caused by the misalignment of an induction rotor shaft were detected using the equipment shown in Figures 2–5. The testing equipment consists of an induction motor, and the details of the instrument are as follows.

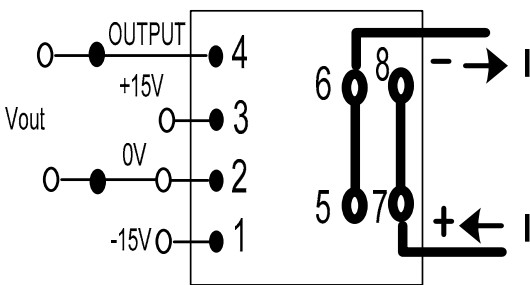

**Figure 2.** The current signal sensors by LEN-HX-10NP [23].

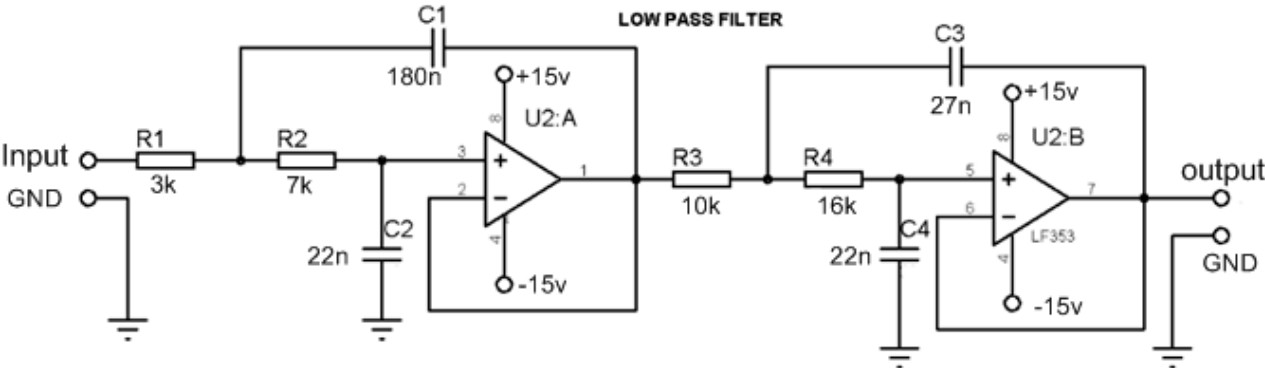

**Figure 3.** Low-pass, filter fourth-order circuit.

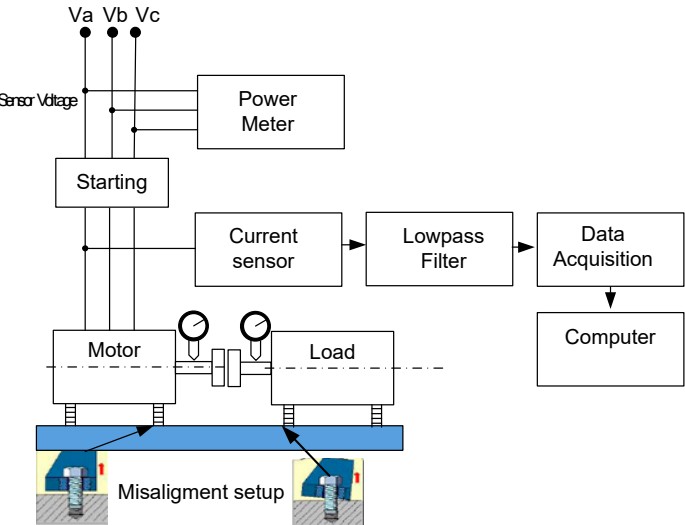

**Figure 4.** Testing diagram of the induction motor and equipment for data recording.

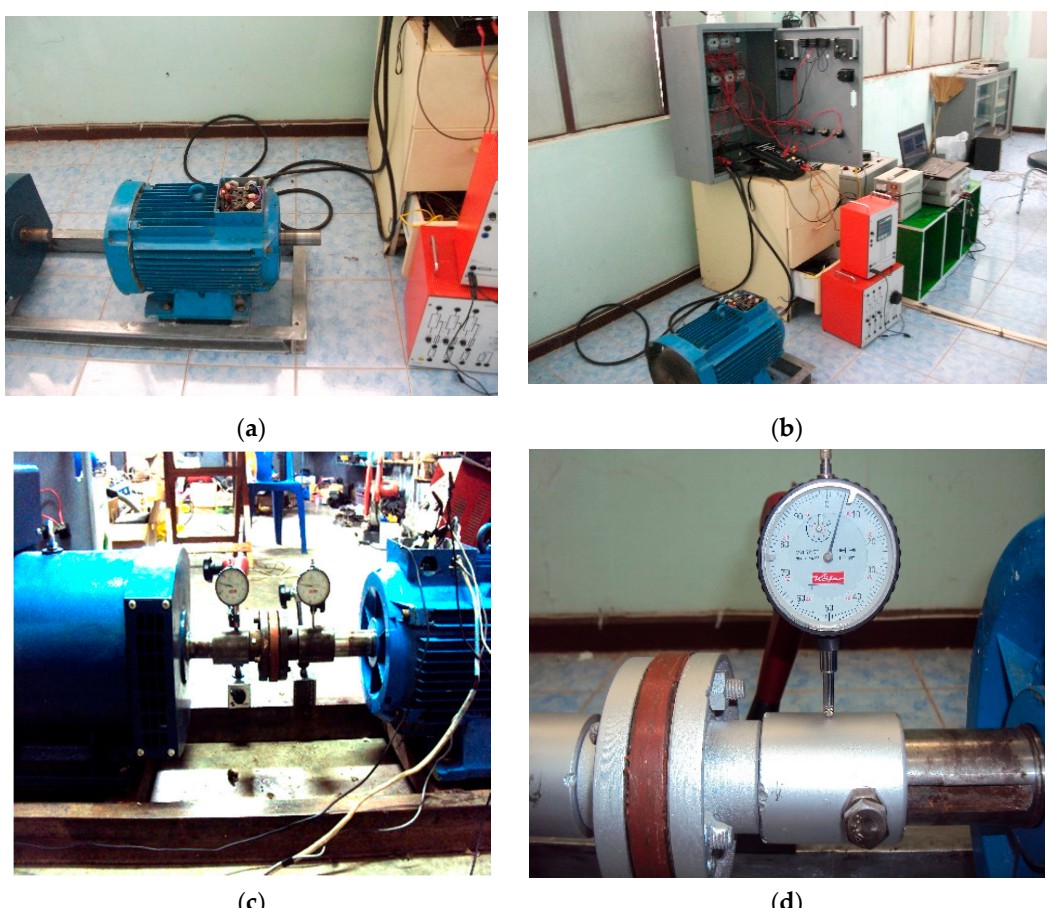

**Figure 5.** (**a**) Induction motor rated 11 kW, (**b**) testing motor with current sensing unit and low-pass filter circuit running without a load connected, (**c**) motor test connected to load, and (**d**) measuring the eccentricity of the rotor shaft with a dial gauge.

1.  The parameters of the three-phase induction motor are 11 kW, 380 V, 50 Hz, and four poles, as shown in Table 1;
2.  The instrument was used for measuring the eccentricity of the rotor shaft;
3.  For the motor startup, the control unit is set to the star and delta switch, which reduces the high current during startup;
4.  The current signal sensors are set with the current type (LEN-HX-10NP), with 1% accuracy, 1% linearity, DC at a 50 kHz frequency bandwidth, 20 A input current Ip, and 4 V output voltage, as shown in Figure 2;
5.  A fourth-order low-pass filter circuit with a cut-off frequency of 500 Hz is included in the design. Because the motor current signal is incorporated with a high frequency, the high-frequency signal needs to be eliminated using the current circuit filter, as shown in Figure 3;
6.  The data acquisition card for receiving signal data for analysis is a Micro USB DAQ that inputs and outputs 30 points and operates in both digital and analog input modes;
7.  Figure 4. shows the circuit that is connected to the equipment for data recording during testing [24]. Figure 5a,b shows the setup of the experimental set with a three-phase induction motor and star-delta starting method. Figure 5c,d shows setup the misalignment installation fault provided by the bolt base under motor, which acts on the base under motor and checking with a dial gauge. The adjustable shaft misalignment by the FISSO Ref: LS30.10 with switch magnet (M); overall height: 367 mm; horizontal: 10 mm dia. × 106 mm length; vertical: 12 mm dia. × 156 mm length; base size: 60 mm × 50 mm × 55 mm; holding strength approx. 800 N; weight: 1.660 kg [25];

8.  The eccentric rotor shaft defect test was carried out as follows. Figure 5a,b shows the test setup for a motor in the no-load mode. A medium motor type was used; therefore, the star-delta starting method was employed to reduce the high current when starting the motor. The low-pass frequency circuit was then detected by sampling at a sampling frequency of 4 kHz, and the current signal was recorded with a DAQ card.

**Table 1.** The parameter of the three-phase induction motor [22].

| Parameter | Value |
|---|---|
| Power | 11 kW |
| Voltage | 380V |
| Ampere | 20A |
| Power factor | 0.8 |
| Rotor speed | 1450 rpm |
| Air gap | 0.5 mm |

Figure 6 shows the process of recording the current signal by the DAQ 6008 card using the LabVIEW program. This process can be used in the analysis of motor faults. As shown in Figure 5c,d, the load was connected to a 5 kW generator, and then four parallel eccentric levels were tested (10%, 20%, 30%, and 40%) using a dial gauge as a level measuring instrument. Then, the stator current signal was detected through a low-frequency circuit. The signal stream was recorded with the DAQ card; the data were recorded using the program LabVIEW. The signal data were obtained in an array form, and the signal was analyzed for the effect of rotor shaft misalignment using a matrix program. To analyze the effects of misalignment faults on the stator current, a test setup was designed in a machinery laboratory. The setup consisted of a motor test stand, which included a dial gauge to set and check the misalignment level, and data collection was performed by sampling the current data and recording the collected data on a PC for analysis. A diagram and photographs of the equipment for the complete test setup are shown in Figures 4 and 5.

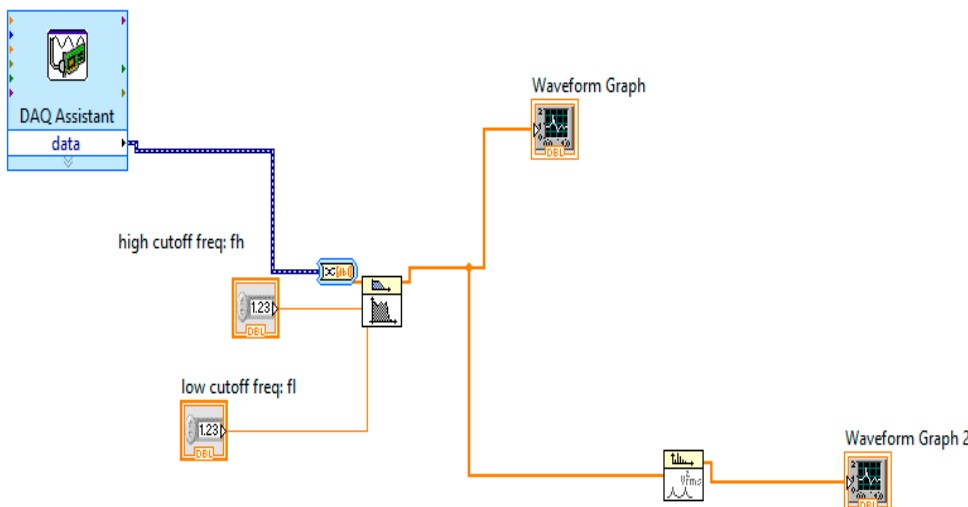

**Figure 6.** Programing using a DAQ 6008 card to record a stator current signal [26].

## 4. Results and Discussion

This paper reports the analysis of an induction motor fault from an incorrectly installed eccentric rotor shaft using current signal detection, FFT, and the discrete wavelet transform (DWT) technique.

### 4.1. Experimental Result at the Normal Condition

The test results for the motor disconnected from the load are as follows.

Results of Current Signal Detection in the Time and Frequency Domains

Figure 7a shows the results of measuring motor current signals in the time domain, whose signal characteristics are non-sinusoidal due to magnetism and hysteresis. Figure 7b shows the signal measurement in the frequency domain using the FFT function; when the motor operates, it produces MMF, resulting in the rotor frequency $fr = sfs$, causing a difference in frequency, and the sideband of the fundamental frequency is obtained for the motor in its normal state. The slip frequency has an energy (dB) value that is lower than the main frequency value fs. The peak (Peak) of the spectrum signal with an odd order (third (150 Hz), fifth (250 Hz), seventh (350 Hz), and ninth (450 Hz)) is obtained, which has a lower energy value than the main frequency.

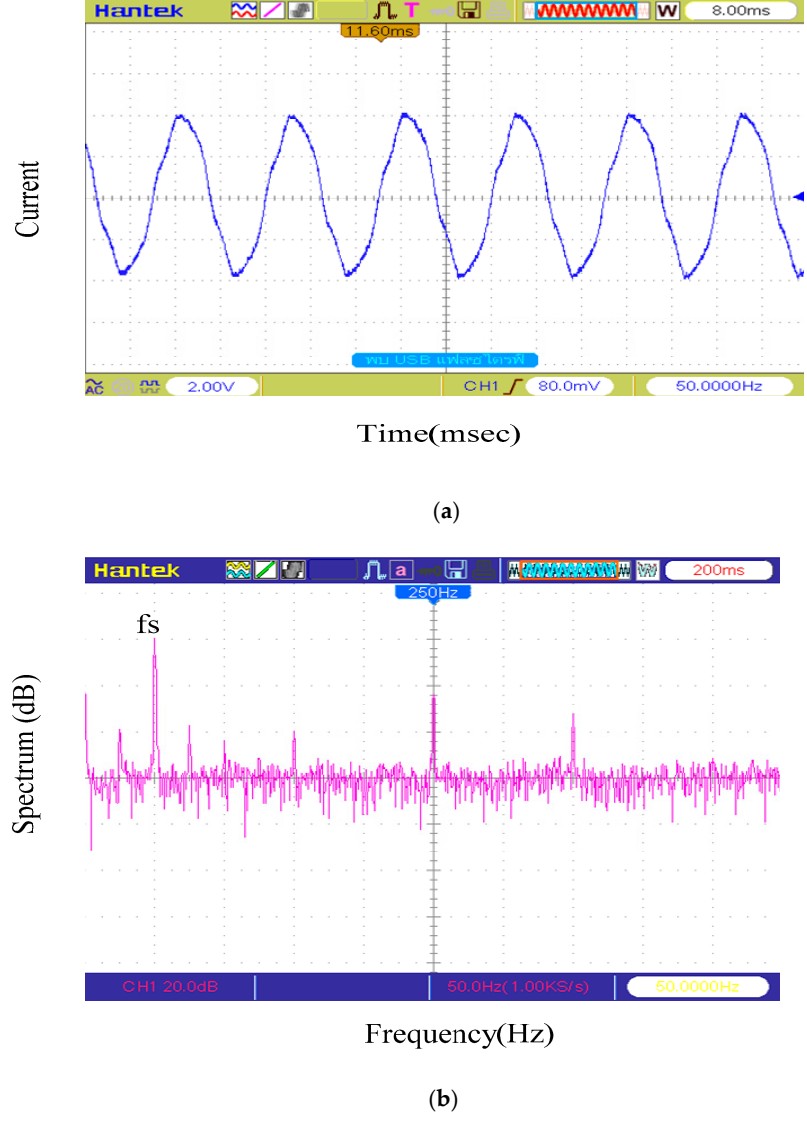

(**a**)

(**b**)

**Figure 7.** The current signal in the time domain and frequency domain motor without load condition. (**a**) shows the results of measuring motor current signals in the time domain, whose signal characteristics are non-sinusoidal due to magnetism and hysteresis. (**b**) shows the signal measurement in the frequency domain using the FFT function; when the motor operates, it produces MMF, resulting in the rotor frequency $fr = sfs$, causing a difference in frequency, and the sideband of the fundamental frequency is obtained for the motor in its normal state. The slip frequency has an energy (dB) value that is lower than the main frequency value fs. The peak (Peak) of the spectrum signal with an odd order (third (150 Hz), fifth (250 Hz), seventh (350 Hz), and ninth (450 Hz)) is obtained, which has a lower energy value than the main frequency.

From Figure 8, by using wavelet decomposition (WD) of the frequency signal, the stator current is divided into four levels including d1, d2, d3, and d4, and one power coefficient out, a4. It can be seen that the harmonic current can appear on both stator and rotor currents. According to Equations (3)–(5), the energy coefficient of stator current can be computed as in Table 2.

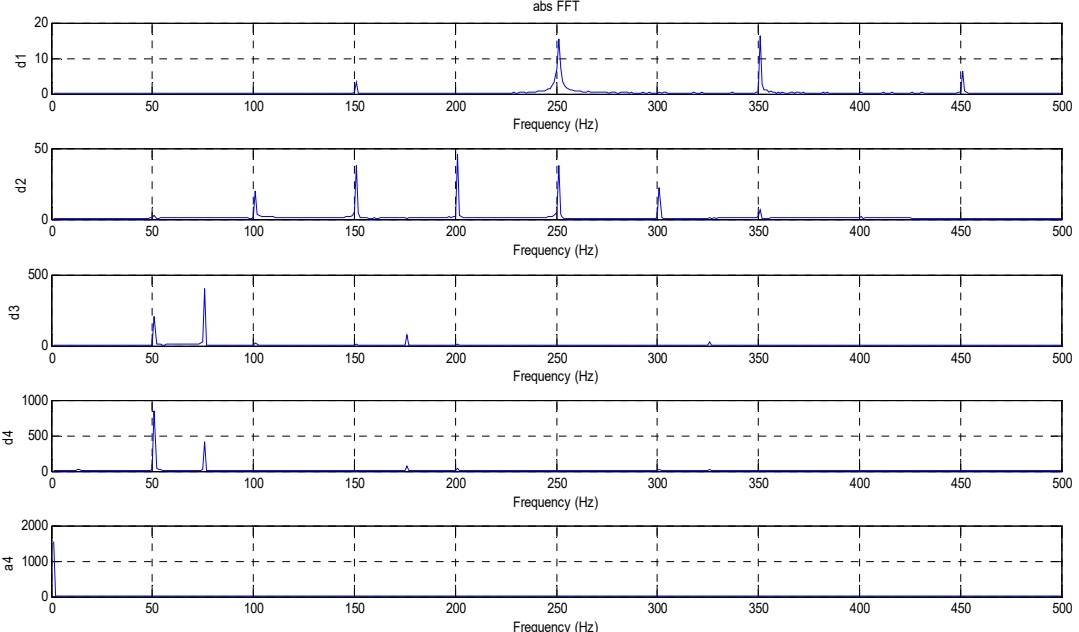

**Figure 8.** Signal current in the frequency domain of the motor; the status is not connected to the load.

**Table 2.** The calculation energy coefficient of stator current signal in normal condition.

| Energy Signal | Coefficient |
| --- | --- |
| Ea | 52.2548 |
| Ed1 | 0.0302 |
| Ed2 | 0.2859 |
| Ed3 | 9.1370 |
| Ed4 | 38.2919 |
| E$_{total}$ | 100 |

Table 2 shows the calculation results for the energy coefficient of the signal in the normal condition, which was 52.2548. The energy coefficient of the current signal was used as a normal standard to compare the normal and fault states.

### 4.2. Analysis Results of the Motor Fault from the Eccentric Rotor Shaft

Figure 5 shows the connection between the motor and load using a 5-kW power generator. The eccentricity was adjusted to four levels—10%, 20%, 30%, and 40%—with a dial gauge. The test results are as follows.

Figure 9a–d shows the test results of the current signal in the frequency domain obtained from the eccentric rotor adjusted at 10%, 20%, 30%, and 40% misalignment levels. The signal energy was analyzed and calculated with the energy coefficient using Equations (3)–(5). The calculation results are as follows.

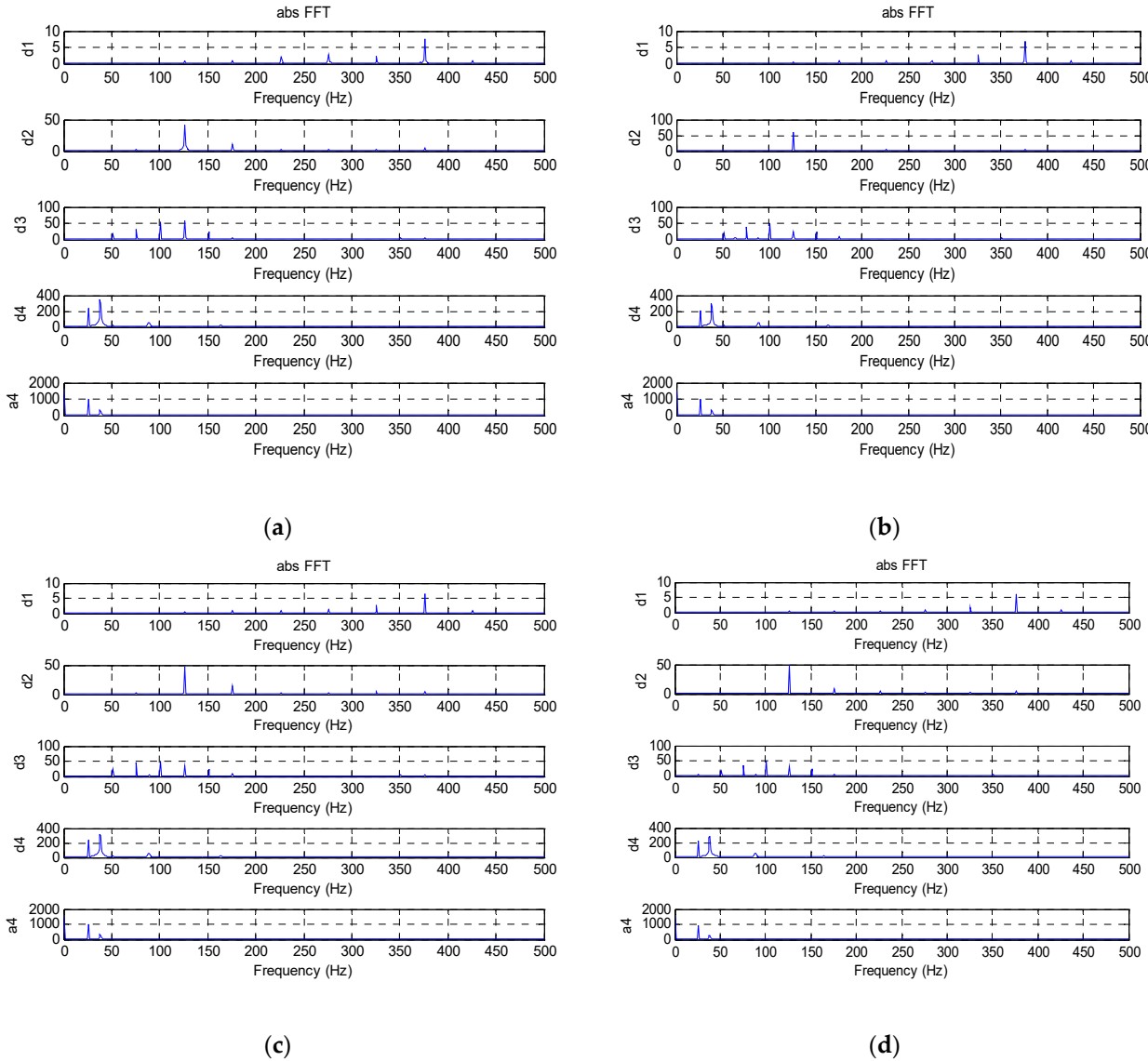

**Figure 9.** Plotting current waveform in frequency domain used wavelet decomposition fourth level. (**a**) misalignment 10% (0.05 mm); (**b**) misalignment 20% (0.1); (**c**) misalignment 30% (0.15 mm); (**d**) misalignment 40% (0.2 mm).

The analysis results of the motor signal current coefficient in Table 3 were used to plot a graph of the linear regression, as shown in Figure 10.

**Table 3.** The analysis results of the motor current signal energy coefficient.

| Parameter Energy Coefficient | Motor Normal Condition | Motor Installation Misalignment | | | |
|---|---|---|---|---|---|
| | **0%** | **10%** | **20%** | **30%** | **40%** |
| $E_a$ | 52.2548 | 90.4908 | 90.5329 | 89.9976 | 89.6416 |
| $E_{d1}$ | 0.0303 | 0.0025 | 0.0025 | 0.0020 | 0.0019 |
| $E_{d2}$ | 0.2859 | 0.1586 | 0.1586 | 0.0927 | 0.1042 |
| $E_{d3}$ | 9.1370 | 0.27690 | 0.2690 | 0.2637 | 0.3154 |
| $E_{d4}$ | 38.2919 | 9.1685 | 9.0370 | 9.6441 | 9.9369 |
| $E_{total}$ | 100 | 100 | 100 | 100 | 100 |

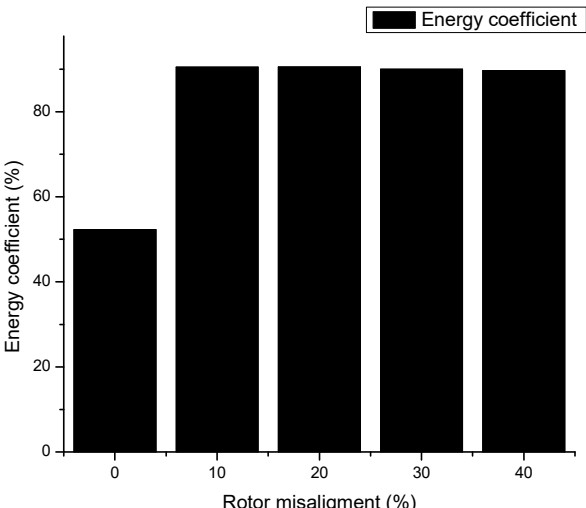

**Figure 10.** The graph of determining the energy coefficient and rotor misalignment.

Figure 10 shows a comparison of the energy coefficients of current signals from the normal motor and the eccentric motor with varying adjustments. The energy coefficient in the normal condition was 52.2548. The eccentric coefficient obtained at 20% misalignment (0.1 mm) was 90.5329; therefore, the impact of the misalignment was 45%. This implies that the slight eccentricity in the installed induction motor connected to a load increases the energy of the stator signal current, which increases the vibration, noise, and temperature. As a result, the useful life of the motor is shorter.

From the linear regression graph in Figure 11, the coefficient of determination was calculated from the linear equation. The correlation coefficient was 0.81162, which indicates a positive relationship between the level of the eccentric rotor and the energy coefficient of the stator current signal at 80.81%. The test and analysis results of the relation reveal the energy coefficient of the current signal, which is an indicator of motor vibration. The same approach was used in the research of Zhang et al. [27], who used vibration data to calculate the frequency band energy ratio to analyze a rotor fault. The results showed that the frequency band ratio could identify a corresponding relationship with the misalignment. Similarly, in our research, statistical processing based on discriminant analysis was applied to identify features in stator current data. However, the selection of the mother wavelet is very important because it affects the fault detection results; hence, it should be carried out carefully.

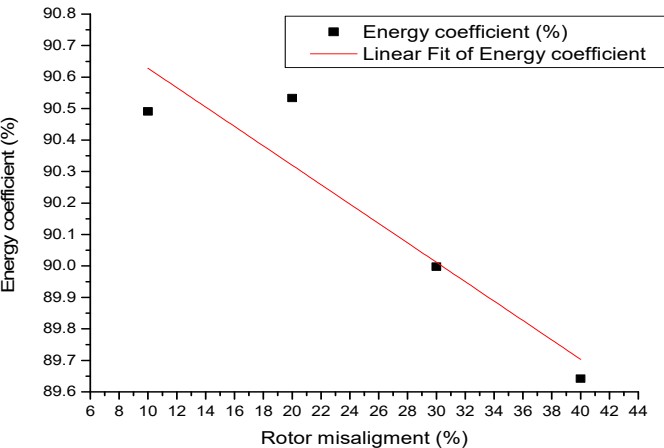

**Figure 11.** The relation between the energy coefficient and the level of the eccentric rotor.

### 5. Conclusions

This article presents a method that applies current signal detection and analysis techniques to diagnose an induction motor fault resulting from installation misalignment of a rotor shaft. FFT and DWT techniques were used to analyze data from a test setup in which external rotor misalignment was the main motor mounting problem. As a result, the vibration increased relative to the degree of misalignment of the shaft. In addition, this affected the outside and inside of the motor, causing an eccentric air gap between the stator and rotor, in addition to changes in magnetic displacement and imbalanced inductance. There was a pair of harmonics (MMF wave causing even harmonics and asymmetry), thus increasing the frequency of the motor current signal (sidebands of the line frequency). The results of the analysis and calculation of the energy coefficient of the signal were correlated. With the increase in the degree of misalignment, it was shown that this analytical technique was effective in analyzing mechanical faults with greater accuracy and detail than a previous method using FFT, and the results agree with those in a previous study [28]. This study expands on past research, with validation from measurement results using more diverse experimental procedures, namely, the measurement of various levels of faults. A limitation of this procedure is that the analysis of signals requires some expertise. Further development of analysis programs should be developed to include automation so that operators can quickly analyze signals to detect faults.

**Author Contributions:** Writing—original draft, S.P.; writing—review & editing, S.T. and S.W. This article presents the collective work of all authors. All authors have read and agreed to the published version of the manuscript.

**Funding:** This research received no external funding.

**Acknowledgments:** The authors would like to acknowledge the support they received from Nakhon Phanom University and Pathumwan Institute of Technology, Thailand.

**Conflicts of Interest:** The authors declare no conflict of interest.

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
