# Peer review of "Testing and Analysis Fault of Induction Motor for Case Study Misalignment Installation Using Current Signal with Energy Coefficient"

_wevj, doi:10.3390/wevj12010037_

Round 1

Reviewer 1 Report

Dear Authors,

The article is dedicated to the verification of an induction motor installation misalignment in the electric drive facility by the original current detection technique via wavelet decomposition.

The actuality of the problem is obvious and the appropriate solution for the on-line diagnosis of the electrical motor misalignment’s installation can prevent electric drives malfunction and extend service life.

It should be noted that many articles have been devoted to the mentioned above problem. Therefore, the first requirement for the authors – to formulate exactly the novelty of their work. The represented in the Introduction explanation should be clarified.

The text has many grammar mistakes – see the attached text of the article with signed remarks (red pen). I showed only part of the errors – detailed editing of the text should be done. The text with my remarks is attached.

All devices used in the research (L.136 for example) should be cited in the references in the appropriate style.

In Fig.6 all included elements must be signed and explained.

L.192-193- text is bot understandable, should be improved.

Expression (6) is not correct, must be fixed.

So long representation of the energy coefficient (Table 3) with 6 digits seems exaggerated.

The representation of the well-known expressions of linear regression and coefficient of determination (eq.7) doesn’t provide significant information and should be eliminated.

Overall impression:

The verification of the submitted method is doubtful:
1. The represented results were obtained only for the no-load mode;
2. The selectivity of the method is questionable since additional factors such as winding faults and others influencing current frequency response were not considered in this work.

The article may be published only after significant improvement considering all represented remarks.

Best wishes,

The reviewer

Author Response

Dear reviewers:

Manuscript ID: Manuscript ID: wevj-1116617 Type of manuscript: Article Title: Testing and Analysis Fault of Induction Motor For Case Study Misalignment Installation Using Current Signal With Energy Coefficient

Authors: supachai prainetr , satean tunyasrirut, santi wangnipparnto*

* Received: 2 February 2021

We are sending a cover letter and revised manuscript.

Thanks for your kindness.

Please address all correspondence to: Supachai Prainetr

We appreciate the kind and generous comments. According to those comments, we revised our paper as following, and we hope that what we had done can meet your requirements.

Thank you so much.

Sincerely yours,

Supachai Prainetr

Dept. of Electrical Engineering

214 Nakhonphanom University, Thailand Tel: +66-81-117-8358

E-mail: prainetr@gmail.com

Reviewer 2 Report

This paper proposed the testing and analysis of faults in an eccentric rotor at different levels using the stator current detection technique and the calculation of the energy signal coefficient via wavelet decom- position (WD) method. And verify its effectiveness through experiments. Suggestions are given in the following areas.

  1. In Section 2.2, the author should not repeatedly explain the same parameters in Formula 1 and Formula 2
  2. In Section 2.3, the parameter formats in formulas 3, 4, and 5 are all normal, and the parameter formats in the parameter introduction are all italics. The author should unify the parameter formats.
  3. The key parameters in Fig. 3 are fuzzy and difficult to see clearly; the parameter values of the abscissa and ordinate in Fig. 9 are too small and difficult to see clearly, please change to a clearer picture.
  4. In Section 2, there are two headings labeled "2.3". Please standardize the application of the headings in the text and carefully check the chapter arrangement in Section 3.
  5. In section 3.1, please give a detailed introduction to formula 6 and explain the meaning of the parameters in the formula.
  6. There are paragraphs with incorrect grammar and unclear sentences in the text, please check carefully.

Author Response

(The authors gave the same response as above.)

Reviewer 3 Report

The paper has a clear and linear structure and is interesting for the journal's readers.

Nevertheless, the presentation needs substantial improvements. In particular, English must be checked throughout the paper. I suggest a native-English colleague proofread the manuscript.

I suggest improving the presentation of the figures, too. Often text size is very small and I don't like the presentation of graphs like in Figures 8 that appears like a screenshot. Some figures appears not significant (as an example figures 4 and 2 (a). If you want to maintain them, please provide more details.

Apices in Equation (6) seem having issues.

In line 269 you must correctly cite the work of Zhang Shu with which you compare the results of your analysis.

Author Response

(The authors gave the same response as above.)

Round 2

Reviewer 1 Report

Dear Authors,

Your manuscript looks much better than the previous version.

However, due to the publication policy of the MDPI journal you should point out the exact type and cite them in the references all used in the experimental tests devices and machines:  L.125 (induction motor); L.131 (current sensor), L.143 (data acquisition), power meter (Figure 4) which is not mentioned in the text,  dial gauges in Figure 5. 

Only after eliminating these flaws, the work can be published. 

Best regards,

The reviewer

Author Response

Dear reviewers:

We appreciate the kind and generous comments. According to those comments, we revised round2 our paper as following, and we hope that what we had done can meet your requirements.

Thank you so much.

Sincerely yours,

Supachai Prainetr

-----------------------------------------------

Reviewer 1: (blue sky highlight)
Comments to the Authors:

Point: However, due to the publication policy of the MDPI journal you should point out the exact type and cite them in the references all used in the experimental tests devices and machines: L.125
(induction motor); L.131 (current sensor), L.143 (data acquisition), power meter (Figure 4) which is not mentioned in the text, dial gauges in Figure 5.

----------------------------------------------------------------------------

Revise, we have cite and reference the experimental test devices as follow.

  • L.127, L.348 (induction motor)
  • L.135, L.351 (current sensor)
  • L.145, L.353 (power meter)
  • L.161, L.355 (dial gauge)
  • L.170, L.357 (DAQ card)
    ------------------------------------------------

Reviewer 2 Report

This paper proposes the use of stator current detection technology to detect and analyze the faults of eccentric rotors at different levels, and use wavelet decomposition to calculate the energy signal efficiency. Experiments show that this method is more effective than fft and is suitable for motor fault analysis and Industrial applications. There are suggestions in the following areas:

  1. In the parameter descriptions of formulas (1) and (2), the format of the parameter subscripts is wrong; there are the same parameters in formulas (1) and (2), please do not introduce these parameters repeatedly.
  2. Please give a more detailed description of Figure 8; the author did not elaborate on the calculation relationship between Figure 8 and Table 2, please add.

Author Response

Dear reviewers

We appreciate the kind and generous comments. According to those comments, we revised round2 our paper as following, and we hope that what we had done can meet your requirements.

Thank you so much.

Sincerely yours,

Supachai Prainetr

-----------------------------------------

Reviewer 2: (green highlight)
Comments to the Authors:

Point: 1. In the parameter descriptions of formulas (1) and (2), the format of the parameter subscripts is wrong; there are the same parameters in formulas (1) and (2), please do not introduce
these parameters repeatedly.
Point: 2. Please give a more detailed description of Figure 8; the author
did not elaborate on the calculation relationship between Figure
8 and Table 2, please add.   

Revise 1, we have revise subscripts and eliminate parameter repeatedly as follow.

  • L.94, L.98

Revise 2, we have add description of Fig.8 in L.213 to L.217.

-----------------------------------------------------------------

Reviewer 3 Report

The text still needs improvement. Many typos are still present. An English-native speaker should check the manuscript.

Some figures can be enlarged.

Author Response

Dear reviewers

We appreciate the kind and generous comments. According to those comments, we revised round2 our paper as following, and we hope that what we had done can meet your requirements.

Thank you so much.

Sincerely yours,

Supachai Prainetr

-----------------------------------------

Reviewer 3:

Comments to the Authors:

Point: The text still needs improvement. Many typos are still present. An English-native speaker should check the manuscript. Some figures can be enlarged.

Revise 1, we have checked manuscripts and has undergone English language editing by MDPI.

Revise 2, we have enlarged figure 9.     

-------------------------------------------
